# Dosimetric Characterization of Small Radiotherapy Electron Beams Collimated by Circular Applicators with the New Microsilicon Detector

Serenella Russo [1,*], Silvia Bettarini [2,*], Barbara Grilli Leonulli [3], Marco Esposito [1], Paolo Alpi [3], Alessandro Ghirelli [1], Raffaella Barca [3], Simona Fondelli [3], Lisa Paoletti [3], Silvia Pini [1] and Silvia Scoccianti [3]

[1] Medical Physics Unit, Azienda USL Toscana Centro, 50012 Florence, Italy; marco1.esposito@uslcentro.toscana.it (M.E.); alessandro.ghirelli@uslcentro.toscana.it (A.G.); silvia.pini@uslcentro.toscana.it (S.P.)

[2] Specialization School in Medical Physics, University of Florence, 50133 Florence, Italy

[3] Radiotherapy Unit, Azienda USL Toscana Centro, 50012 Florence, Italy; barbara.grillileonulli@uslcentro.toscana.it (B.G.L.); paolo.alpi@uslcentro.toscana.it (P.A.); raffaella.barca@uslcentro.toscana.it (R.B.); simona.fondelli@uslcentro.toscana.it (S.F.); lisa.paoletti@uslcentro.toscana.it (L.P.); silvia.scoccianti@uslcentro.toscana.it (S.S.)

[*] Correspondence: serenella.russo@uslcentro.toscana.it (S.R.); silvia.bettarini@unifi.it (S.B.)

**Featured Application: A complete and accurate dose distribution characterization of small radiation therapy electron beams collimated by circular applicators is performed by the new generation silicon detector before starting the clinical use.**

**Abstract:** High-energy small electron beams, generated by linear accelerators, are used for radiotherapy of localized superficial tumours. The aim of the present study is to assess the dosimetric performance under small radiation therapy electron beams of the novel PTW microSilicon detector compared to other available dosimeters. Relative dose measurements of circular fields with 20, 30, 40, and 50 mm aperture diameters were performed for electron beams generated by an Elekta Synergy linac, with energy between 4 and 12 MeV. Percentage depth dose, transverse profiles, and output factors, normalized to the $10 \times 10 \ cm^2$ reference field, were measured. All dosimetric data were collected in a PTW MP3 motorized water phantom, at SSD of 100 cm, by using the novel PTW microSilicon detector. The PTW diode E and the PTW microDiamond were also used in all beam apertures for benchmarking. Data for the biggest field size were also measured by the PTW Advanced Markus ionization chamber. Measurements performed by the microSilicon are in good agreement with the reference values for all the tubular applicators and beam energies within the stated uncertainties. This confirms the reliability of the microSilicon detector for relative dosimetry of small radiation therapy electron beams collimated by circular applicators.

**Keywords:** radiotherapy electron beams; output factors; tubular applicator; solid-state detector



## 1. Introduction

Precise radiation doses to localized tumours can be delivered by electron beam radiotherapy due to their rapid dose fall-off and short range, allowing the treatment of targets close to the surface, while sparing the underlying tissues. High-energy electron beams generated by linear accelerators, with energy between 4 and 12 MeV, are used. Small electron beams, suitable for the treatment of limited lesions, can be shaped to the target region by using stainless steel tubular applicators fastened to the accelerator head with small aperture diameter varying from 20 to 50 mm. A complete and accurate dose distribution characterization of small radiation therapy electron beams, collimated by these circular applicators, is mandatory before starting the clinical use.

However, the dosimetry of small and/or irregular electron fields is challenging. Electron beam side deflections, due to multiple Coulomb scattering, occur in small circular collimators, resulting in reduced output and, therefore, to a lack of lateral-scatter equilibrium. This fact plays an important role in small electron beams dosimetry: the depth of maximum dose moves toward the surface, the central axis percentage depth dose decreases, and the output dose/monitor unit decreases when electron field sizes are smaller than or comparable to the radius required for lateral scattering equilibrium [1,2]. Moreover, the size of the detector, with respect to the field size, has to be considered. The detector volume averaging effect, which is dependent on detector size, affects small electron beam dose measurements as well as in narrow photon beams.

Therefore, a detector with high spatial resolution capability and high sensitivity is required, which is also tissue equivalent. Previous studies have largely investigated various parameters such as detector type and volume that affect the dosimetry of photon beams in small fields, leading to the recommendation of the TRS483 on small field dosimetry [3]. However, little attention has been given in literature to small electron beams [2,4,5].

Plane–parallel ionization chambers may be not adequate for small electron beam dosimetry because of their too large collecting electrode and consequent beam non-uniformity over the chamber area. Both silicon diodes and diamond dosimeters were found to be suitable to perform relative electron dosimetry [1,6,7], including the Output Factors (OFs) determination. However, only a couple of studies reported dosimetric data for beams shaped by tubular applicators [4,5].

The novel PTW (PTW-Freiburg, Germany) microSilicon detector may be a suitable detector for measurements in high-energy small-field electron beams. The performance of this small-volume detector, for dosimetry in small photon beams, was already studied, showing very good dosimetric properties [8–10]. In addition, a three-dimensional characterization of the active volumes of the PTW microSilicon has been performed for accurate Monte Carlo simulations of the detector [11]. Recently, this detector was characterized for electron beam dosimetry of the reference 10 cm × 10 cm field [12]. However, to our knowledge, the dosimetric performances of the novel PTW microSilicon detector, under small electron beams, have not been assessed. The aim of this work is the dosimetric characterization of clinical electron beams from 4 to 12 MeV, generated by a linear accelerator and shaped by commercial tubular applicators with the novel PTW microSilicon detector, by comparison with commercially available dosimeters.

## 2. Materials and Methods

The percentage depth-dose (PDD) curves, transverse beam profiles, and output factor (OF) measurements of small-sized circular fields were performed for 4 to 12 MeV nominal energy range of electron beams, generated by an Elekta Synergy linear accelerator (https://www.elekta.com/radiotherapy/treatment-delivery-systems/elekta-synergy/, accessed on 10 December 2021). The percentage depth dose (PDD) is defined as the ratio, expressed as a percentage, of the absorbed dose response at a depth d ($D_d$) to the absorbed dose response at the depth, corresponding to the maximum dose R100 ($D_{R100}$) along the central axis of the beam [1], according to the formula:

$$\text{PDD} = \frac{D_d}{D_{R100}} \times 100. \tag{1}$$

Electron beams were shaped by stainless steel tubular applicators, which consist of a main circular part fastened to the accelerator head and a set of add-on field defining tubes with 20, 30, 40, and 50 mm aperture diameter, at a distance of 95 cm from the source (SSD). All dosimetric data were collected during the same measurement session in a PTW MP3 motorized water phantom, at SSD of 100 cm, using three different solid-state detectors: a novel PTW microSilicon detector (model No. 60023), a PTW diode E (model No. 60017), and a PTW microDiamond (model No. 60019). Each detector was used in parallel configuration (i.e., with the detector axis parallel to the beam axis).

The plane–parallel chamber PTW Advanced Markus was used as reference dosimeter in the 50 mm diameter circular field and depth dose curves, measured by diode E, microDiamond, and microSilicon detectors and compared to dose distributions measured by the ion chamber to assess their suitability [13].

### 2.1. Detectors

The PTW microSilicon is a new unshielded p-type silicon diode detector, recommended for measurements in small electron and photon beams due to its very small sizes (radius = 0.75 mm, thickness = 0.18 μm). The disk-shaped sensitive volume is 0.03 mm$^3$, and the reference point position is on the chamber axis, 0.9 mm from its tip, which also corresponds to the effective point of measurement for photons. The effective point of measurement for electron beams is located at 0.3 mm from the chamber tip. No bias voltage needs to be applied.

The PTW microDiamond is a synthetic single crystal diamond Schottky diode, with a disk-shaped sensitive volume of 2.2 mm in diameter and about 1 μm thick. The sensitive volume is located below the detector surface at a water equivalent depth of 1 mm. The device operates with no external bias voltage applied. Pre-irradiation with a dose equal to 5 Gy was performed.

The PTW diode E is a previous generation silicon diode detector, characterized by a circular sensitive volume of a 1 mm$^2$ circular and 30 μm thickness. The effective point of measurement is located 0.8 mm below the detector front surface, and the water-equivalent window is 1.33 mm thick. No bias voltage was applied.

The depth ionization distributions, measured by the PTW Advanced Markus, were converted to depth dose distributions by multiplying the ionization charge at each measurement depth by the stopping-power ratio sw$_{air}$ at that depth, according to the IAEA TRS-398 protocol [14]. The variation of ion recombination and polarity effects with depth was also considered.

### 2.2. Percentage Depth-Dose Curves and Beam Profiles

PDD curves were measured along the beam central axis for the 4, 6, 9, and 12 MeV beam energies and for all field sizes, with the three solid-state detectors and the plane-parallel ionization chamber only for the 50 mm diameter field. The alignment of each detector, with respect to the central axis, was initially performed with the PTW Trufix positioning system and then carefully verified by profile scanning both in-plane (i.e., in the gun–target direction) and cross-plane (i.e., perpendicular to the gun–target direction) directions for all the investigated field sizes and for each beam energy.

The effective point of measurement was considered for PDD measurements performed by all the detectors. The scan direction was toward the surface to reduce the effect of meniscus formation, and a step size of 0.4 mm was used for the entire range.

No correction was applied to the PDD curves for the diamond nor for the silicon detectors, since the water-to-carbon and water-to-silicon mass collision stopping power ratios are nearly constant in the range 1–20 MeV [14].

Transverse beam profiles (in-plane and cross-plane) measurements were performed for each energy and beam aperture, with all the three solid-state dosimeters at zref determined by PDD measurements for the normalization field 10 cm × 10 cm. We decided to measure all beam profiles at the same depth for a better comparison of profile parameters since field size and penumbra values are very sensitive to the measurement depth, PDDs and transverse profiles were acquired and analysed by the PTW TANDEM electrometer and the PTW Mephysto MC$^2$ software, respectively. No reference detector was used for all dosimetric curve measurements, but a step size of 0.4 mm was used for the entire range. The normalization of all profiles to the central axis was performed.

### 2.3. Output Factors

The Output Factor is the ratio of dosimeter readings, under a given set of non-reference conditions, to that measured under reference conditions [14]. OFs measurements were evaluated for each tubular applicator and all beam energies by the three solid state detectors. The $10 \times 10$ cm$^2$ field size was used for the normalization of OFs. A PTW plane–parallel ionization chamber, Advanced Markus, was also used only for the tubular applicator with 50 mm aperture diameter.

Each detector was initially centred with the PTW Trufix positioning system in the 3D water scanner. By adopting the methodology recommended in TRS483 [3] for small photon beam dosimetry, two orthogonal profiles were subsequently acquired at the reference depth, for each beam energy with the smallest field size, by using the three solid state detectors. Then, each detector was positioned at the maximum detector signal point. A step size of 0.2 mm was used for the entire range to maximize the positioning accuracy.

Output factors were evaluated as the absorbed dose at zmax for the field of interest (20, 30, 40, and 50 mm aperture diameter of the tubular applicators) relative to the absorbed dose at zref for the $10 \times 10$ cm$^2$ normalization field, at SSD of 100 cm, and with the same number of monitor units as recommended by the TRS398 [14]. For detectors such as diodes and diamonds, the output factor was obtained as the ratio of the detector reading under the actual beam size, relative to the normalization field. When the ionization chamber is used, the variation of water-to-air stopping-power ratio with depth was accounted for in the OF determination. A PTW Unidos E Universal Dosimeter was used, and measurements were repeated five times for each field size with 100 monitor units (corresponding to about 1 Gy) for each readout.

### 2.4. Data Analysis

The main parameters obtained from the PDD curves were R100, R80, and R50, i.e., the depths in water of the corresponding maximum value of the absorbed dose (following the definition of PDD in Section 2 and in Equation (1)) of the 80%, and of the half of its maximum value, respectively. The practical range Rp (only for the 50 mm diameter beam size) and the mean energy at the phantom surface $E_0$ were also determined.

The difference over the entire depth range and the difference in distance (mm) within the region of the curve maximum slope (20—80%) were evaluated to compare PDDs for all beam energies: for the 50 mm diameter tubular applicator, the differences between each solid-state dosimeter and the ionization chamber were evaluated, while for the remaining tubular collimators, the diode E was chosen as the reference detector.

The profiles measured by the different solid-state detectors were compared in terms of field size, evaluated by the full-width half maximum (FWHM, in mm), and in terms of the penumbra, defined as the distance in mm between the points, corresponding to the 20% and the 80% of the maximum dose along the average beam profile (averaging is performed between the left- and right-side values of the profile).

Similarly to the PDD case, the difference over the entire off-axis range and the difference in distance (mm) within the penumbra region were evaluated between the profiles measured by the diode E and the other two solid state detectors, the microDiamond and the microSilicon, respectively.

The maximum uncertainty, related to detector positioning over the PDD curve and dose profiles, was calculated by multiplying the used scanning step size (0.4 mm) for the evaluated dose gradient in different regions of the dose curve.

For OFs evaluation, the reference detector choice was dissimilar according to the tubular applicator size: for the 50 mm aperture diameter, the ionization chamber was considered as the reference, while for the other beam diameters (20, 30, and 40 mm), the diode E was considered as the reference. Relative differences between OF values, measured by each detector and the reference one, were calculated.

Four main components of uncertainty were considered for OF determination due to: the position accuracy of the detector, the electrometer reading, the statistical dispersion

of repeated measurements, and the reproducibility of the OF values between different measurement sessions.

Comparisons of dose parameters, measured by the reference detector and other dosimeters, were performed using a two-tailed Wilcoxon signed-rank test, considering a *p*-value of <0.05 as significant.

## 3. Results

### 3.1. Percentage Depth-Dose Curves and Beam Profiles Measurements

Figure 1 shows the PDD curves measured for all beam energies with the three solid-state detectors and the plane-parallel ionization chamber for the 50 mm diameter field. The differences between PDDs measured by the solid-state dosimeters and the ionization chamber are also shown. An agreement of ±3% was observed among the detector responses for all the beam energies over the whole PDD curve, except for the region close to the water surface, where the detectors are partially in air. The difference, grown to about 5% in the maximum dose gradient area, is only for the 12 MeV energy beam. The maximum difference in distance (mm) within the 20–80% region, reported in Table 1, for each solid-state detector relative to the reference data, was 0.7 mm for all beam energies, except for 12 MeV, where 1.3 mm is the maximum distance. As far as concerned only the diode E, the maximum difference in distance was 0.4 mm in all curves. No statistically significant differences were found between the reference data and those measured by the other detectors.

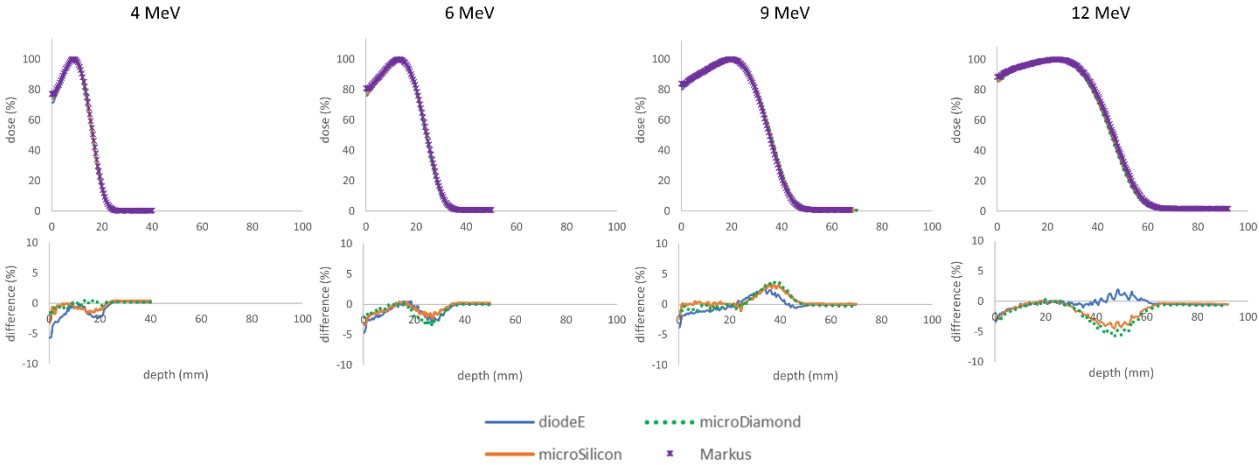

**Figure 1.** In the upper part: PDD curves measured with the three solid-state detectors and the plane-parallel ionization chamber for the 50 mm diameter field; the percentage doses are relative to the peak of the PDD curve. In the bottom part: % differences between PDDs, measured by the solid-state dosimeters and the ionization chamber.

**Table 1.** Maximum difference in distance (mm), within the 20-80% PDD region for the solid-state detectors (diode E reported as d-E, microDiamond as m-D and microSilicon as m-Si), with respect to the chosen reference for each beam energy. The mean value over all beam energies was also shown.

| | Difference in Distance (mm) | | | | | | | | |
|---|---|---|---|---|---|---|---|---|---|
| | **20 mm** | | **30 mm** | | **40 mm** | | | **50 mm** | |
| **Energy (MeV)** | **m-D** | **m-Si** | **m-D** | **m-Si** | **m-D** | **m-Si** | **d-E** | **m-D** | **m-Si** |
| 4 | 0.3 | 0.4 | 0.2 | 0.1 | 0.3 | 0.4 | 0.3 | 0.03 | 0.1 |
| 6 | 0.2 | 0.1 | 0.3 | 0.1 | 0.4 | 0.1 | 0.4 | 0.4 | 0.3 |
| 9 | 0.4 | 0.9 | 0.4 | 0.4 | 0.5 | 0.7 | 0.4 | 0.7 | 0.6 |
| 12 | 0.5 | 0.8 | 1.1 | 0.4 | 1.1 | 0.6 | 0.3 | 1.3 | 0.9 |
| mean | 0.4 | 0.6 | 0.5 | 0.3 | 0.6 | 0.5 | 0.4 | 0.6 | 0.5 |

For the remaining tubular collimators, the maximum difference over the entire depth range was within 4%, and the maximum difference in distance within the region of the curve maximum slope was 1.1 mm for all beam energies. Similar results were obtained for the microDiamond and microSilicon detectors. The PDD curves, measured by the three solid-state detectors in the 20 mm diameter field for all beam energies, are reported in Figure 2, together with the respective percentage differences between PDDs, measured by the microDiamond, the microSilicon dosimeters, and the diode E data.

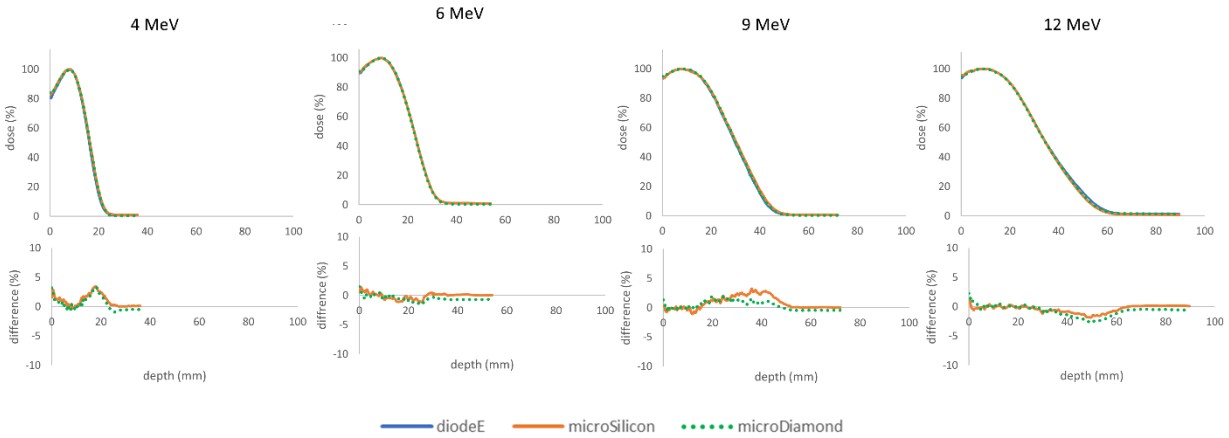

**Figure 2.** PDD curves measured for the 20 mm diameter field with the solid-state detectors: % differences between microSilicon and microDiamond data, and diode E reference measurements were also shown.

Table 2 summarizes the most relevant dosimetric parameters extracted from PDD curves for the 50 mm diameter field, showing a maximum deviation of about 1 mm for all the investigated depth parameters and 0.3 MeV for $E_0$, measured by the solid-state detectors, with respect to the plane-parallel ionization chamber.

**Table 2.** Most relevant dosimetric parameters, extracted from PDD curves for the 50 mm diameter field, measured by ionization chamber Markus (M), diode E (d-E), microDiamond (m-D), and microSilicon (m-Si).

| | | | | | | | | Diameter 50 mm | | | | | | | | | | | |
|---|---|---|---|---|---|---|---|---|---|---|---|---|---|---|---|---|---|---|---|
| Energy (MeV) | R100 (mm) | | | | R80 (mm) | | | | R50 (mm) | | | | Rp (mm) | | | | $E_0$ (MeV) | | |
| | M | d-E | m-D | m-Si | M | d-E | m-D | m-Si | M | d-E | m-D | m-Si | M | d-E | m-D | m-Si | M | d-E | m-D | m-Si |
| 4 | 8.4 | 9.2 | 8.8 | 8.8 | 13.4 | 13.4 | 13.3 | 13.5 | 16.5 | 16.3 | 16.3 | 16.5 | 21.1 | 20.8 | 20.9 | 21.2 | 3.8 | 3.8 | 3.8 | 3.8 |
| 6 | 13.2 | 12.8 | 13.6 | 12.8 | 20.2 | 20.1 | 20.0 | 19.8 | 24.4 | 24.1 | 24.1 | 24.0 | 30.8 | 30.1 | 30.5 | 30.2 | 5.7 | 5.6 | 5.6 | 5.6 |
| 9 | 19.6 | 20.0 | 19.6 | 19.6 | 29.5 | 29.9 | 29.8 | 29.9 | 35.1 | 35.4 | 35.6 | 35.6 | 43.5 | 43.3 | 44.0 | 44.1 | 8.2 | 8.3 | 8.3 | 8.3 |
| 12 | 23.2 | 22.8 | 24.0 | 224 | 38.9 | 38.9 | 38.1 | 37.9 | 46.6 | 46.7 | 45.7 | 45.4 | 57.0 | 58.1 | 56.4 | 56.1 | 10.9 | 10.9 | 10.6 | 10.6 |

The dosimetric characteristics of all applicators, except for the maximum diameter field, and all beam energies are reported in Table 3: a maximum deviation of 1.6 mm on R100, of 0.8 mm for all the other depth parameters investigated, and 0.2 MeV for $E_0$ were found between microDiamond and microSilicon measurements and the diode E data. The electron beam parameters R100, R80, and R50 shifted toward the phantom surface with decreasing field dimensions, particularly for the smallest field size. Furthermore, these parameters increased with increasing energy, as electrons penetrated deeper into the phantom. $E_0$ was quite constant for 4 and 6 MeV beam energies and increased with the field size for the remaining beam energies, with a maximum deviation of 32% from the nominal value, of the highest energy value of 12 MeV, for the smallest field size.

**Table 3.** Most relevant dosimetric parameters derived from PDD curves for all tubular applicators, measured by the solid-state dosimeters diode E (d-E), microDiamond (m-D), and microSilicon (m-Si).

| | | | | | | | | | | | | |
|---|---|---|---|---|---|---|---|---|---|---|---|---|
| **Diameter 20 mm** | | | | | | | | | | | | |
| | **R100 (mm)** | | | **R80 (mm)** | | | **R 50 (mm)** | | | **E$_0$ (MeV)** | | |
| **Energy (MeV)** | **d-E** | **m-D** | **m-Si** | **d-E** | **m-D** | **m-Si** | **d-E** | **m-D** | **m-Si** | **d-E** | **m-D** | **m-Si** |
| 4 | 8.4 | 8.0 | 8.0 | 13.0 | 13.2 | 13.1 | 16.2 | 16.5 | 16.4 | 4.4 | 4.5 | 4.5 |
| 6 | 9.6 | 9.2 | 8.8 | 17.4 | 17.2 | 17.3 | 22.7 | 22.6 | 22.5 | 5.3 | 5.3 | 5.2 |
| 9 | 7.6 | 7.2 | 7.6 | 21.3 | 21.6 | 21.7 | 29.7 | 30.4 | 30.0 | 6.9 | 7.1 | 7.0 |
| 12 | 10.0 | 8.5 | 8.8 | 24.5 | 24.5 | 24.4 | 34.9 | 34.8 | 34.6 | 8.1 | 8.1 | 8.1 |
| **Diameter 30 mm** | | | | | | | | | | | | |
| | **R100 (mm)** | | | **R80 (mm)** | | | **R 50 (mm)** | | | **E$_0$ (MeV)** | | |
| **Energy (MeV)** | **d-E** | **m-D** | **m-Si** | **d-E** | **m-D** | **m-Si** | **d-E** | **m-D** | **m-Si** | **d-E** | **m-D** | **m-Si** |
| 4 | 8.6 | 8.6 | 8.8 | 13.3 | 13.3 | 13.3 | 16.3 | 16.4 | 16.4 | 4.4 | 4.4 | 4.4 |
| 6 | 12.0 | 12.0 | 11.6 | 19.5 | 19.5 | 19.3 | 24.1 | 23.9 | 23.7 | 5.6 | 5.6 | 5.5 |
| 9 | 14.4 | 13.6 | 14.0 | 27.0 | 26.6 | 26.5 | 34.1 | 34.0 | 33.8 | 7.9 | 7.9 | 7.9 |
| 12 | 16.8 | 15.6 | 15.6 | 32.0 | 32.1 | 31.9 | 42.3 | 42.1 | 41.7 | 9.9 | 9.8 | 9.7 |
| **Diameter 40 mm** | | | | | | | | | | | | |
| | **R100 (mm)** | | | **R80 (mm)** | | | **R 50 (mm)** | | | **E$_0$ (MeV)** | | |
| **Energy (MeV)** | **d-E** | **m-D** | **m-Si** | **d-E** | **m-D** | **m-Si** | **d-E** | **m-D** | **m-Si** | **d-E** | **m-D** | **m-Si** |
| 4 | 9.2 | 8.8 | 9.2 | 13.6 | 13.9 | 13.7 | 16.5 | 16.8 | 16.7 | 3.8 | 3.9 | 3.9 |
| 6 | 13.6 | 13.6 | 12.0 | 20.1 | 20.0 | 19.8 | 24.2 | 24.2 | 23.9 | 5.6 | 5.6 | 5.6 |
| 9 | 19.6 | 18.8 | 20.0 | 29.5 | 29.5 | 29.5 | 35.4 | 35.8 | 35.6 | 8.2 | 8.3 | 8.3 |
| 12 | 19.5 | 18.0 | 19.4 | 36.3 | 36.2 | 36.0 | 45.1 | 44.7 | 44.3 | 10.5 | 10.4 | 10.3 |

The crossline profiles, measured by the different solid-state detectors, and the difference in % over the entire off-axis range, between microDiamond and microSilicon measurements and the diode E data for the smallest circular applicator, were shown in Figure 3. Both for in-plane and cross-plane profiles, for all tubular applicators and beam energies, the maximum difference in % over the entire range was within 1% in the low gradient regions and grew up to 3% in the maximum slope area. Furthermore, in these regions, the maximum difference, in distance, was within 0.4 mm. No statistically significant differences were found between the reference data and microDiamond or microSilicon results.

In Table 4, the field size, evaluated by the FWHM, and the 80–20% penumbra values for all applicators and beam energies were reported: a maximum deviation of 0.4 mm in field size and penumbra evaluation was found for both microDiamond and microSilicon, with respect to the diode E, without a significant difference between the two detectors.

### 3.2. OF Measurements

The OF values, obtained using all the dosimeters for all the tubular applicators and beam energies, are shown in Figure 4. OF values were reported as a function of nominal field size, defined as the applicator diameter. OFs measured by the solid-state detectors agreed with those measured with the ionization chamber within 1.7% for the 50 mm diameter, as can be seen in Table 5. It is noteworthy that the microSilicon got the best agreement with the ionization chamber, with a maximum deviation of 0.8%.

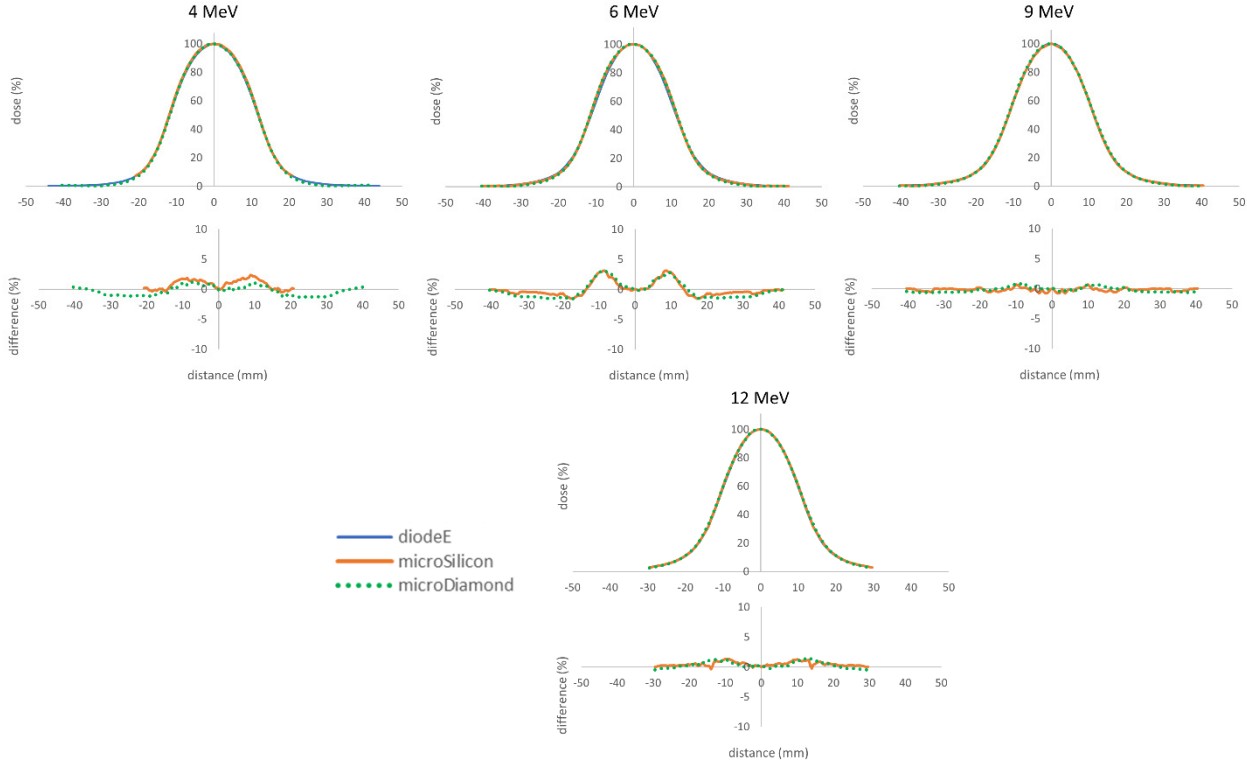

**Figure 3.** Crossline beam profiles, measured by the different solid-state detectors, and % respective difference over the entire off-axis range between microDiamond and microSilicon measurements and diode E data.

**Table 4.** Field size and 80-20% penumbra values for all circular applicators, measured by the solid-state dosimeters diode E (d-E), microDiamond (m-D), and microSilicon (m-Si).

| Field Size (mm) | | | | | | | | | | | | |
|---|---|---|---|---|---|---|---|---|---|---|---|---|
| **Field Diameter (mm)** | **4 MeV** | | | **6 MeV** | | | **9 MeV** | | | **12 MeV** | | |
| | **d-E** | **m-D** | **m-Si** | **d-E** | **m-D** | **m-Si** | **d-E** | **m-D** | **m-Si** | **d-E** | **m-D** | **m-Si** |
| 50 | 54.3 | 54.3 | 54.5 | 53.5 | 53.6 | 53.7 | 52.8 | 52.8 | 53.0 | 52.9 | 52.9 | 53.0 |
| 40 | 43.4 | 43.7 | 43.6 | 42.7 | 43.0 | 43.1 | 42.3 | 42.5 | 42.4 | 42.2 | 42.5 | 42.6 |
| 30 | 32.9 | 33.1 | 33.2 | 32.9 | 32.5 | 32.5 | 31.8 | 32.0 | 32.0 | 31.7 | 32.0 | 32.0 |
| 20 | 23.4 | 23.6 | 23.8 | 22.9 | 23.2 | 23.1 | 22.7 | 22.8 | 22.6 | 22.4 | 22.7 | 22.7 |
| Penumbra (mm) | | | | | | | | | | | | |
| **Field Diameter (mm)** | **4 MeV** | | | **6 MeV** | | | **9 MeV** | | | **12 MeV** | | |
| | **d-E** | **m-D** | **m-Si** | **d-E** | **m-D** | **m-Si** | **d-E** | **m-D** | **m-Si** | **d-E** | **m-D** | **m-Si** |
| 50 | 11.5 | 11.4 | 11.4 | 11.6 | 11.8 | 11.8 | 12.3 | 12.4 | 12.0 | 12.0 | 12.5 | 12.3 |
| 40 | 10.9 | 10.9 | 10.8 | 11.0 | 11.0 | 10.8 | 11.5 | 11.6 | 11.3 | 11.6 | 11.9 | 11.6 |
| 30 | 10.4 | 10.1 | 10.1 | 10.4 | 10.2 | 10.2 | 10.9 | 10.7 | 10.7 | 10.9 | 11.1 | 11.1 |
| 20 | 8.9 | 8.6 | 8.6 | 9.0 | 8.8 | 8.8 | 9.5 | 9.4 | 9.4 | 9.6 | 9.8 | 9.7 |

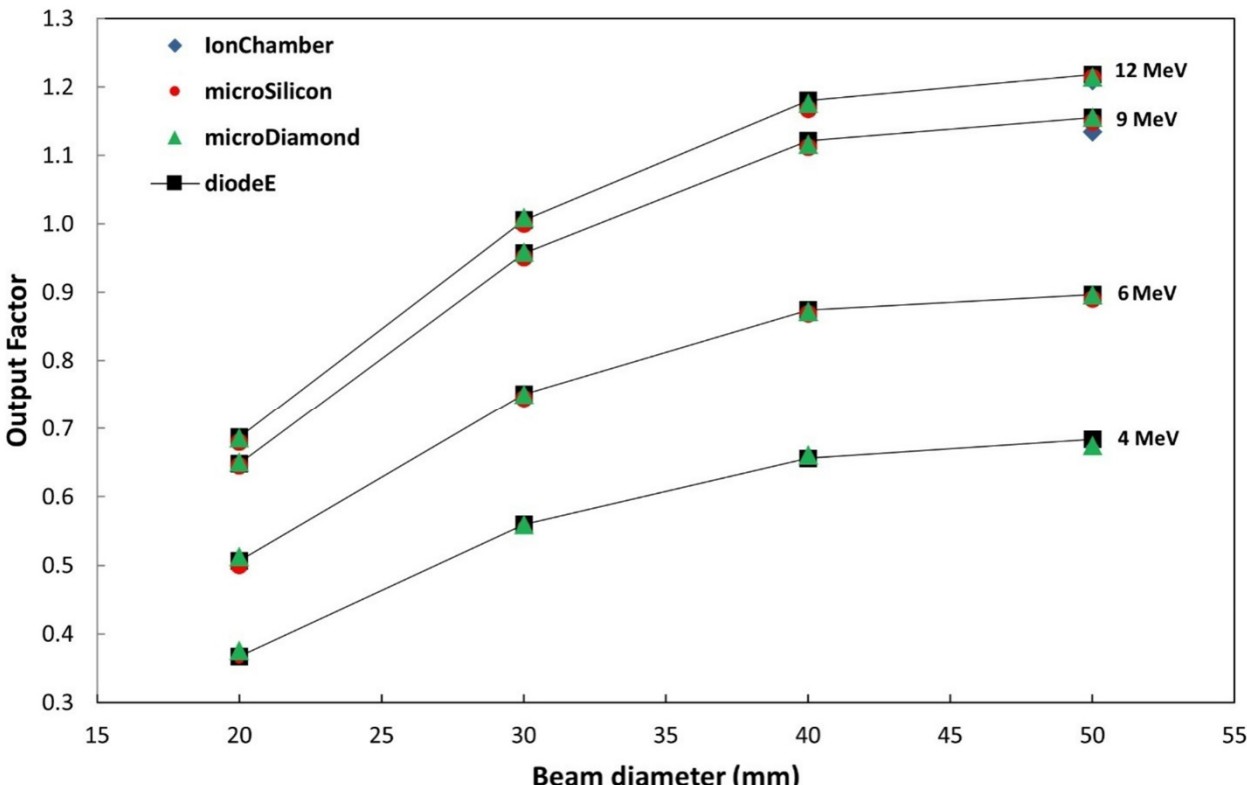

**Figure 4.** Output factor values, measured by all dosimeters, for all aperture diameters and beam energies. Measurement uncertainties were within the sizes of point indicators.

**Table 5.** OF values measured for all beam energies with all the dosimeters for the 50 mm diameter tubular applicator (ionization chamber Markus, reported as M, diode E as D-E, microDiamond as m-D, and microSilicon as m-Si) and only with solid-state dosimeters for the remaining applicators.

| Beam Diameter (mm) | Dosimeter | 4 MeV OF | 4 MeV Deviation from Ref. (%) | 6 MeV OF | 6 MeV Deviation from Ref. (%) | 9 MeV OF | 9 MeV Deviation from Ref. (%) | 12 MeV OF | 12 MeV Deviation from Ref. (%) |
|---|---|---|---|---|---|---|---|---|---|
| 20 | d-E | **0.367** | - | **0.506** | - | **0.648** | - | **0.687** | - |
|  | m-D | 0.376 | 2.5 | 0.513 | 1.3 | 0.650 | 0.3 | 0.686 | 0.1 |
|  | m-Si | 0.364 | 0.8 | 0.500 | 1.2 | 0.645 | 0.5 | 0.680 | 1.0 |
| 30 | d-E | **0.560** | - | **0.751** | - | **0.957** | - | **1.005** | - |
|  | m-D | 0.560 | 0.0 | 0.751 | 0.0 | 0.959 | 0.2 | 1.010 | 0.4 |
|  | m-Si | 0.557 | 0.6 | 0.744 | 0.9 | 0.960 | 0.3 | 0.999 | 0.6 |
| 40 | d-E | **0.656** | - | **0.874** | - | **1.121** | - | **1.180** | - |
|  | m-D | 0.661 | 0.7 | 0.873 | 0.2 | 1.117 | 0.4 | 1.176 | 0.3 |
|  | m-Si | 0.657 | 0.2 | 0.869 | 0.6 | 1.111 | 0.9 | 1.167 | 1.1 |
| 50 | M | **0.672** | - | **0.889** | - | **1.140** | - | **1.205** | - |
|  | d-E | 0.684 | 1.7 | 0.897 | 0.8 | 1.155 | 1.3 | 1.218 | 1.1 |
|  | m-D | 0.675 | 0.4 | 0.897 | 0.9 | 1.156 | 1.4 | 1.215 | 0.8 |
|  | m-Si | 0.672 | 0.1 | 0.890 | 0.1 | 1.149 | 0.8 | 1.213 | 0.6 |

For the 20, 30, and 40 mm applicator diameters, OFs agreement between the microSilicon data and the reference values was always within 1.2%. All OF values measured by microDiamond agreed with the reference data: differences were typically within 1.4%, with a maximum value of 2.5% for 4 MeV and the smallest aperture.

### 3.3. Uncertainty Budget Evaluation

The maximum uncertainty, related to detector positioning over the whole PDD curve, was estimated to be in the range 2–4% for all the beam energies and field sizes. This finding was related to the product of the used scanning step size of 0.4 mm for the evaluated dose gradient of about 4% mm$^{-1}$ for the higher beam energy and 9% mm$^{-1}$ for the lower beam energy, regardless of the field size.

For all the profile curves, the maximum uncertainty related to detector positioning was estimated to be within 0.4% in the centre area and 3% in the maximum slope area, for all beam energies and field sizes, given the step size of 0.4 mm. Moreover, another component of the uncertainties of about 1% must be introduced to take into account that the reference detector was not used for relative dosimetry.

For OF determination, the positioning accuracy resulted to be ±0.2 mm, and therefore, the uncertainty related to detector positioning was lower than 0.1%. This finding takes into account the method used to identify the centre field in the present study [15]. The accuracy of the electrometer was ±0.2%, according to the manufacturer. The measurement readout uncertainties were calculated as one standard deviation of the mean of repeated measurements and resulted lower than 0.5% for solid-state detectors and 0.2% for the ionization chamber. The reproducibility of the OF values, obtained in different measurement sessions, was about 1% (1 SD) for the solid-state dosimeters and about 0.5% (1 SD) for the ionization chamber. By combining the components with Gaussian error propagation, the total uncertainties for the measurements were estimated to be about 1.5% for the solid-state dosimeters and 1% for the ionization chamber.

### 4. Discussion

The electron disequilibrium and high-gradient dosing make the small electron beam dosimetry challenging. Diode detectors are considered suitable for measurements of the scanning data and the relative output factors in small electron beams [13]. The microSilicon detector has been recently introduced to overcome the performance of its predecessor diode E because of its smaller sensitivity to temperature, higher dose stability and lower dose per pulse dependence [16]. In fact, the microSilicon entrance window is more water equivalent, thanks to the reduction in density of the casting compound on top of the silicon chip, compared to the predecessor [11]. The lack of investigation of the microSilicon detector performance in high-energy small electron beams boosted the need for the characterization of clinical electron beams shaped by commercial tubular applicators performed in this study.

According to IAEA 398, plane-parallel air-filled ionization chambers are recommended to be used for PDD measurements at all electron energies and, below 10 MeV, their use is mandatory. However, the diameter of the collecting electrode should be considered when performing small beam dosimetry to avoid the influence of radial non-uniformities of the beam profile [14]. TG25 suggested that measurements can be done with an ionization chamber that is small enough such that its active volume fits in the flat portion of the beam [1,13]. Diode and diamond detectors can be used as an alternative to the ionization chamber after verifying that the detector is suitable for depth dose measurements trough test comparisons at a representative beam quality [14].

To this purpose, the Advanced Markus ionization chamber has been used as reference for PDD and OF measurements for the 50 mm diameter applicator, and the differences between each solid-state dosimeter and the ionization chamber were evaluated. Diode E detector presented the overall minimum difference in dose and distance with the ion chamber data, and silicon diode detectors have been considered suitable for electron dosimetry for a long time in the literature [1,13]. So far, the diode E was chosen as the reference detector for the remaining tubular collimators to compare measurements performed by the microSilicon and the microDiamond.

The dose difference, in % over the entire depth range, was evaluated for PDD. However, the dose difference may not be appropriate to compare the variability among detectors in the region between the R80 and the R20, where the dose gradient is higher. So, the

distance-to-agreement, representing the minimum distance to the same value from the reference data was calculated for the solid-state detectors.

We found that differences between solid-state detectors are still consistent with the measurement uncertainty, taking into account the step size used for data acquisition and the absence of the reference detectors due to the small size of the investigated fields.

The results reported in Table 1, regarding the maximum distance-to-agreement, averaged over all beam energies between the ion chamber and each solid-state dosimeter data, were found to be similar to the ones presented by Akino et al. [12]. In particular, for the 50 mm tubular applicator, our results were 0.4 mm, 0.6 mm, and 0.5 mm for diode E, microDiamond, and microSilicon, respectively. Similarly, for the same solid-state detectors, Akino et al. [12] obtained 0.51 mm, 0.68 mm, and 0.86 mm, respectively, in the $10 \times 10$ cm$^2$ reference field size.

A maximum deviation of 0.4 mm in penumbra evaluation was found in our measurements. In this study transverse beam profiles measurements were performed for each energy and beam aperture at zref, as derived for the reference field $10 \times 10$ cm$^2$. Differently, Bagalà et al. acquired beam profiles at the depth of maximum dose, as derived from PDD measurements for different detectors, and they found a maximum difference up to 1.1 mm in penumbra values for beam profiles measured at slightly different depths [4].

As expected, the electron beam parameters R100, R80, and R50 shifted toward the phantom surface with decreasing field size for a fixed beam energy, while E$_0$ increased with the field size for the higher beam energies. This is due to the multiple scattering occurring in smaller circular applicators, resulting in a slight decrease in Eo due to the energy loss in the collision with the collimator lateral side.

A decrease in the OF value is observed for each field size by decreasing the beam energy and for each energy by reducing the field size. Accordingly, the OF for the smallest field size and the lower energy was about three times lower than the reference field size. OFs larger than unity were observed at 9 MeV and 12 MeV, with an increase for the 50 mm diameter field size at 12 MeV up to 20% in comparison with the output of the $10 \times 10$ cm$^2$ field size. The same behaviour was observed by Di Venanzio et al. [5] with an increase for the 50 mm diameter field size at 15 MeV up to 30%.

OF determined in this study show a moderate to high difference with the values measured by Venanzio et al. [5]: for all field sizes at 6 MeV beam energy deviations were within 3%, but an increase up to about 30% was observed for 12 MeV beam energy and the 20 mm beam diameter. This confirms that different components determine the linac output when tubular applicators are used. Therefore, dosimetric measurements should be performed for any configuration similar to patient irradiation, using reliable high-resolution detectors [5].

## 5. Conclusions

Beam output in small electron beams, shaped by circular applicators, is influenced by electron scattering from applicator walls, air enclosed by an applicator, and scattering in the water. Dosimetric parameters for these small fields are peculiar to each configuration, and their measurement for each field size is recommended. The selection of a suitable dosimeter for the measurement of electron parameters, in small fields with lateral scatter disequilibrium, is a critical issue. Moreover, the methodology used for accurate measurements of dose curves and beam output is also a fundamental aspect in the dosimetry of small electron beams.

In this study, dose profiles PDDs and OFs, measured by the PTW microSilicon detector, were found to be in good agreement with reference detector measurements for all the tubular applicators and beam energies, within the stated uncertainties. This confirms that PTW microSilicon is a suitable detector for the dosimetry of small electron beams collimated with circular applicators and can be considered as a reliable alternative to its predecessor diode E.

**Author Contributions:** Conceptualization, S.R. and S.B.; methodology, S.R.; software, S.B.; validation, B.G.L., R.B., L.P., S.S., P.A. and S.F.; formal analysis, S.B.; investigation, S.B.; data curation, S.R. and S.B.; writing—original draft preparation, S.R., S.B.; writing—review and editing, M.E., S.P., A.G.; supervision, S.S. All authors have read and agreed to the published version of the manuscript.

**Funding:** This research received no external funding.

**Conflicts of Interest:** The authors declare no conflict of interest.

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
