# Peer review of "Dosimetric Characterization of Small Radiotherapy Electron Beams Collimated by Circular Applicators with the New Microsilicon Detector"

_applsci, doi:10.3390/app12020600_

Round 1
Reviewer 1 Report
An excellent manuscript. I would recommend it for publication without changes.
Author Response
We would like to thank the reviewer for the positive comments on our work.
Reviewer 2 Report
The manuscript presents very interesting research with convincing results to apply novel high-resolution detectors of the electron beam radiation for therapy. There are no significant remarks.
There is only some thinking around such type of scientific paper. Not all readers understand specific terms such as Output Factor and its physical mean. Perhaps, it corresponds to factor for absolute dose recalculation, perhaps no. Time of irradiation? There is no information about the level of maximum measured dose and dynamic range of detectors. At highest beam energies (12 MeV), is there some induced radioactivity? And what about additional electron deceleration emission of X-ray radiation/bremsstrahlung? It can be much higher percentage than declared accuracy of direct beam measurements.
Also it’s reliable to clear some abbreviations such as PTW (in each page), “… at SSD of 100 cm” (line 148).
The manuscript can be published without additional reviewing.
Author Response
We thank the reviewer for the valuable comments and observations. Please see the attachment for the detailed answers.

Reviewer 3 Report
Thanks for sharing your work. Please find attached my comments in a .pdf file.

Author Response
We thank the reviewer for the valuable comments and observation. Please see the attachment to find the point-to-point answers and the consequent changes made on the manuscript.

Round 2
Reviewer 3 Report
Thanks for responding to my comments and your amendments to the manuscript.